# From Cone to Seed and Seedling—Characterization of Three Portuguese *Pinus pinaster* Aiton Populations

**Paula Maia** * and **Sofia Corticeiro** *

Biology Department, University of Aveiro, and CESAM, Campus Universitário de Santiago,
3810-193 Aveiro, Portugal
* Correspondence: paula.maia@ua.pt (P.M.); sofiacorticeiro@ua.pt (S.C.)

**Abstract:** Seed production in *Pinus pinaster* Aiton is not usually considered a limiting factor for natural regeneration; instead, seed weight is a more limiting factor in successful pine recruitment. Divergent relationships between seed weight and germination rate were previously observed amongst maritime pine populations of central coastal Portugal. The present study followed cone-to-seed and seed-to-seedling approaches to decrease intra-specific variability and clarify the impact of cone size and seed mass on seedling recruitment. The main objectives of this study were (1) to determine and compare the mass of cones and seeds of three maritime pine populations located along a geographic gradient along the coastal center of Portugal and (2) to clarify the relations thereof between cone and seed traits with germination phenology and initial seedling growth. Results demonstrated that heavier cones tended to generate more mature seeds, but not necessarily heavier ones, although seed weight was suggested to be an indicator of robust seedlings. The outcomes of this study reinforce the great intra-population variability of maritime pine, showing the ecology of this species and its ability to adapt to various environments successfully.

**Keywords:** maritime pine; cone mass; seed mass; germination; recruitment; seedling growth

## 1. Introduction

In the Mediterranean region, natural regeneration has long been a major concern in sustainable forest management [1,2], often affected by silvicultural practices, environmental constraints, climate and/or natural disturbances, such as wildfires [3]. Natural regeneration allows the maintenance of the local adaptative genetic pool [4], constituting an advantage to species to better cope with abiotic and biotic stresses during seed germination and the early stages of seedlings, increasing the chances of successful recruitment [5]. Understanding the processes and the main drivers within natural regeneration will help predict the main limitations and adequate forest management practices to improve the natural regeneration success of forest species following a disturbance.

Maritime pine (*Pinus pinaster* Aiton) is an obligate seeder depending on the viability of the canopy seed bank to regenerate after a disturbance [6–8]. Maritime pine includes diverse ecotypes thoroughly adapted to a wide range of edaphic and climatic factors with substantial intraspecific phenotypic and genetic variations [9–12] related to the great diversity of ecological environments in which they evolved. Differences in growth and reproductive strategies among maritime pine populations have gained special relevancy in the last years [4,13–17] due to the continuous decrease in maritime pine forest area as a consequence of frequent wildfires and land-use changes [18,19].

Cone production is one of the features highly variable among and within maritime pine populations [11,12], with considerable additional fluctuations related to climate conditions (temperature and precipitation), particularly during the period of seed formation prior to cone maturation [20]. Seed production in this species is not usually considered a limiting factor for natural regeneration [3]. One of the most limiting traits in maritime

pine for the success of recruitment is instead seed weight [3,4,21–25]. Seed weight is not a relatively constant value, with a variation range within populations equal to or even higher than the mean variations found between different populations [21]. Seed mass is markedly affected by the maternal environment, with maritime pine trees from non-stressed environments presenting larger cones, larger seeds and less variability in seed weight within the population [21,22,26].

The decrease in maritime pine forest area during the last years in Portugal [27] has highlighted the need to accurately predict the recruitment of maritime pine Portuguese populations in order to be able to develop strategies to overcome the challenges that climate changes are bringing to the sustainability of the maritime pine forest. The maritime pine forest in the Central Coastal area represents a significant part of the maritime pine area in Portugal, highly relevant in terms of local provisioning, regulating and cultural ecosystem services [28–32]. The characterization of these populations in terms of the dynamic between seed production and seedling performance during the initial growth stages is crucial for improving the success of natural regeneration under global climate change scenarios. This is a topic that gains even more relevance when considering the heterogeneous combinations of soil, climate and ecological conditions found in the Portuguese forest, in which these maritime pine populations have evolved [28–32].

In the current study, cone-to-seed and seed-to-seedling approaches were followed to clarify the impact that cone and seed mass have on seedling germination and early development under controlled conditions. Under this context, the main objectives of this study were (1) to determine and compare the mass of cones and seeds of maritime pine populations of three populations of the central coastal region of Portugal and (2) to clarify the relations thereof between cone and seed traits with germination phenology and initial seedling growth, crucial stages for the recruitment success of maritime pine.

## 2. Materials and Methods

### 2.1. Study Site Characterization

The study area is comprised of maritime pine stands selected from three populations located along a geographic gradient within central coastal Portugal (Figure 1): Mira (M), Tocha (T) and Praia da Vieira (PV). These same-age populations are located along a geographic gradient along the coastal center of Portugal, being established in similar conditions, considering climate, soil type, altitude, and mean distance to the sea. The selected maritime pine populations are similarly aged, approximately 40 years old, with no evidence of recent management operations or fires. According to the Köppen classification [33], this region is included in the Mediterranean Csb climate zone, a temperate climate with dry and mild summers. The mean annual precipitation varies between 822 to 945 mm.year$^{-1}$, with mean temperatures ranging from 13.3 to 15.5 °C [33]. The lithology of the pine stands is mainly constituted of dunes and sand dunes, which are coarse-textured [30,32,34], with a mean elevation of 15 to 21 m and a mean distance to the sea of between 1.5 and 3.4 km.

### 2.2. Climate Data

Annual climate data of the selected sites was obtained from IPMA (years 2019, 2018, 2017) [35–37] for the two hydrological years prior to seed sampling. According to the ecology of maritime pine, these two years correspond to the period of cone formation and seed production and maturation. The climate variables analyzed were annual temperature (average, maximum, and minimum), annual precipitation, and maximum consecutive days without rain. Data were analyzed by hydrological year: November 2017 to October 2018 (year 1) and from November 2018 to October 2019 (year 2).

### 2.3. Cone Sampling and Processing

In the spring of 2019, mature two-year-old cones were collected from the 3 selected populations described previously. For each population, one mature two-year-old cone was collected per tree on 30 trees. Cones were dried for 2 to 3 days in a forced air circulation oven

at 45 °C to melt the resin seal closing the scales and release the seeds without damage, which were subsequently individually stored in paper bags. Immediately after cone opening, seeds were manually extracted, and the cone was weighed. The total number of seeds per cone was determined.

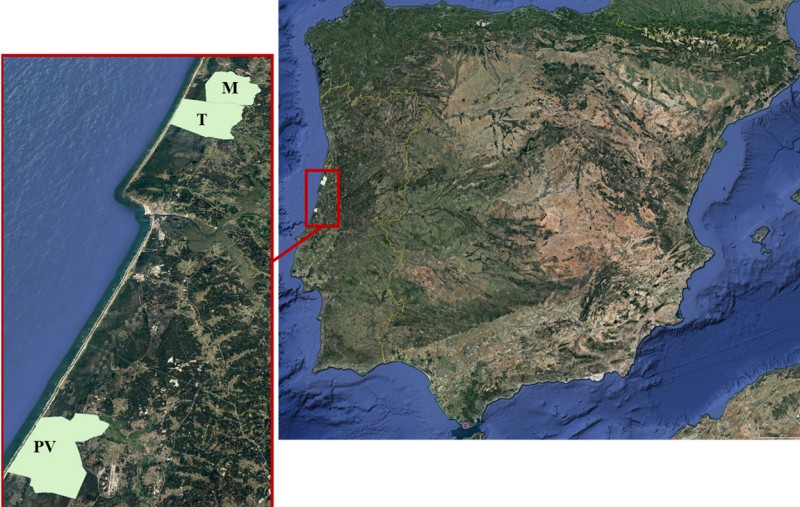

**Figure 1.** Maritime pine selected populations Mira (M), Tocha (T) and Praia da Vieira (PV), distributed along the North-to-South geographic gradient within the study area.

*2.4. Seed Germination*

Seed germination rate was evaluated by testing 16 seeds randomly selected per cone on a total of 30 cones by population. A total of 480 seeds were used to characterize each population. Individual seeds were soaked in deionized water for 16 h (overnight) to discard floating seeds after having been individually weighed. Soaked seeds were individually placed on trays with pre-marked cells, grouped by cone, so all the individual seeds could be followed throughout the germination trial. The trays were lined with moist filter paper, and, when necessary, the filter paper was re-moistened with deionized water. Seeds were left to germinate in a laboratory at room temperature (24/18 °C day/night temperatures) under a 14 h–10 h photoperiod for 3 months. Germinated seeds (radicle > 1 mm) were removed daily. At the end of the experiment period, the germination rate of each population was calculated by cone. The cone germination potential, computed as the number of seeds per cone multiplied by the germination rate, was assessed for each population. Seed cumulative germination was also computed by population as the proportion of the total number of seeds that had germinated throughout the various weeks of the experiment. The proportion of floating seeds, which were not considered in the germination experiment, was negligible in all populations, not affecting the final number of seeds tested.

*2.5. Seedling Nursing and Biometric Traits*

Immediately after germination, each of the seeds removed from the germination tray was placed into a 300 mL plastic pot filled with a volumetric proportion of a 2:1 peat: perlite mixture. The pots with the germinated seeds were placed in a greenhouse at room temperature (24/18 °C day/night temperatures) under a 14 h–10 h photoperiod. Seedlings were watered to field capacity once a week and left to grow for a period of 7 months. Once a month, the seedlings were supplied with a standard NPK nutritional solution (12:4:6). At the age of 7 months, the total height, the stem height, and the basal shoot diameter of seedlings were measured. Total height and stem height were measured from the base of the stem. Total height included the length of the last whorl of leaves, while stem height was measured until the base of the canopy, defined as the first green leaves.

*2.6. Statistical Analysis*

Differences in seed germination percentage (yes/no) and germination rate over time among the 3 maritime pine populations (Mira (M), Tocha (T) and Praia da Vieira (PV)) were analyzed by performing generalized linear models (GLM), following a binomial error distribution using the logit link function and a Poisson distribution with the log link function, respectively. The fitness of the models was assessed using the Hosmer and Lemeshow goodness of fit test. Cone mass, number of seeds per cone, seed mass, cone germination potential and seedling biometric variables were analyzed through one-way ANOVA, followed by the post hoc Tukey HSH test. Spearman correlations were used to determine the relation between (1) cone mass and the number of seeds produced, (2) number of seeds per cone and mean seed mass and (3) cone mass and mean seed mass. All data analyses were carried out with R software, version 4.1.1. using the packages stats (4.1.1), ggpubr (0.4.0), and dplyr (1.0.8).

**3. Results**

The annual precipitation was higher in year 1 for all populations varying from 952 mm/year in PV to 1115 mm/year in M. In year 2, however, the annual precipitation was inferior; this tendency was similar between populations, with M presenting the highest value of 960 mm/year, and PV presenting the lower value of 742 mm/year. Although there was a higher amount of precipitation during year 1, the maximum number of consecutive days without rain within that year was 98 in M and T and 100 in PV (Table 1). During year 2, the maximum sequential days with no record of precipitation varied from 30, in M and T, to 35 in PV. Regarding the mean annual temperature (Table 1), no significant differences were detected between years or populations. On the other hand, year 1 reached higher maximum temperatures (39.9 °C to 40.8 °C) when compared to year 2 (34.9 °C to 35.8 °C) in all populations. The minimum temperature was recorded in M during year 1, the only day when the temperature reached below zero ($-0.9$ °C).

**Table 1.** Climate data of the selected sites Mira (M), Tocha (T) and Praia da Vieira (PV) for year 1 (November 2017 to October 2018) and for year 2 (November 2018 to October 2019). The climate information included the annual precipitation (Precip), the mean annual temperature (Mean Temp), the maximum (Temp max) and the minimum (Temp min) temperatures recorded within each year and the maximum consecutive days without rain (Max days without rain). Error ($\pm$) indicates the standard deviation in Mean Temperature.

| Site | Year | Precip (mm/year) | Mean Temp (°C) | Temp Max (°C) | Temp Min (°C) | Max Days without Rain |
|------|------|------------------|----------------|----------------|----------------|-----------------------|
| M | 1 | 1115 | $15.3 \pm 4.6$ | 40.8 | $-0.9$ | 98 |
| | 2 | 960 | $15.5 \pm 3.9$ | 35.8 | 0.1 | 30 |
| T | 1 | 1082 | $15.2 \pm 4.5$ | 40.5 | 0.2 | 98 |
| | 2 | 838 | $15.6 \pm 3.9$ | 35.4 | 1.0 | 30 |
| PV | 1 | 952 | $15.4 \pm 4.7$ | 39.9 | 1.6 | 100 |
| | 2 | 742 | $15.7 \pm 4.1$ | 34.9 | 2.2 | 35 |

Cone mass was not significantly different ($p > 0.05$) among populations due to the high variability of cone sizes found within each population. Cone mass varied from 41.8 g in PV to 152.9 g in M (Table 2).

The average number of seeds by cone (Table 2) varied significantly between M and the other two populations ($\chi^2 = 422.780$, $p < 0.05$), with cones from M having the highest average seed value in a cone (136). T had the highest variability within a population, with the number of seeds found in a cone ranging from 20 to 180. Mean seed mass was significantly different ($\chi^2 = 31.518$, $p < 0.05$) amongst populations, although highly variable, particularly in the T population, being 0.055 g in PV, 0.057 g in M and 0.059 g in T.

The mass of the empty cone (without seeds) had a strong positive correlation with the number of seeds produced ($\rho = 0.579$, $p < 0.05$). A weak but positive significant correlation

(ρ = 0.32, *p* < 0.05) was identified between pine cone mass and mean seed mass, suggesting that heavier cones tend to originate heavier seeds. No significant correlation was found between the number of seeds per cone and the total mass of those seeds (ρ = 0.03, *p* > 0.05).

**Table 2.** Mean cone mass, the mean number of seeds per cone, and mean seed mass of maritime pine populations from Mira (M), Tocha (T) and Praia da Vieira (PV). Error (±) is the standard deviation. Different letters indicate statistically significant differences (*p* < 0.05) between populations as detected by pairwise Mann–Whitney U-test.

| Population | Cone Mass (g) | Seeds Per Cone | Mean Seed Mass (g) | Germination Rate (%) |
|---|---|---|---|---|
| M | 100.3 ± 26.9 [a] | 135.7 ± 30.4 [a] | 0.056 ± 0.01 [a] | 76.7 [b] |
| T | 89.1 ± 28.6 [b] | 80.4 ± 43.5 [b] | 0.059 ± 0.01 [b] | 88.9 [a] |
| PV | 88.6 ± 26.8 [b] | 116.3 ± 27.5 [c] | 0.054 ± 0.01 [c] | 72.5 [b] |

The seed germination experiment lasted for three months, when it could be assured that the remaining seeds would not germinate. The seed germination profile (Figure 2) varied between populations. The first seeds germinated seven days after the beginning of the experiment from the T population. After 30 days, the germination rate was about 22% in T, 28% in M and 33% in PV, while after 60 days, the germinated rate was around 52% in PV, 58% in T and 64% in M. The final germination rate was achieved at day 81 for all populations, with values of 72% in PV, 78% in M and 89% in T, with significant differences ($\chi^2$ = 10.015, *p* > 0.05). The seed germination rate at the end of the experiment was significantly different between T and the other 2 populations, being significantly positively related to the mean seed mass and mean cone mass (Table 3).

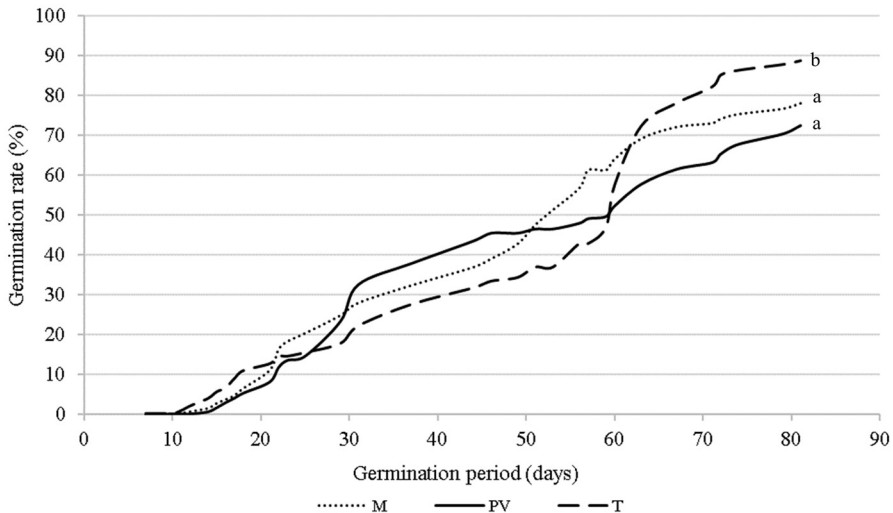

**Figure 2.** Germination (%) over time of maritime pine seeds from Mira (M), Tocha (T) and Praia da Vieira (PV) populations. Different letters indicate significant differences (*p* < 0.05) in seed germination over time.

The seed germination rate increased with seed mass; that tendency was even more pronounced in T than in M and PV. Although there were differences in seed mass between M and PV, no significant differences were found between the germination rate of those two maritime pine populations. By analyzing differences in cone mass, it was possible to verify that even though the significant differences were detected, the influence that this variable has on the germination rate of the three populations is neglectable. In order to translate the germination rate of M, T and PV populations into potential maritime pine recruitment, the cone germination potential was investigated. The cone germination potential can be described as an indicator of the capability of a single cone to originate future new seedlings, considering the germination rate within each population and the

average number of seeds produced per cone. This parameter does not intend to predict maritime pine recruitment in the field; it is only intended to compare the probability of maritime pine regeneration amongst the population considering those variables. The cone germination potential, although highly variable within each population, was significantly different among populations ($\chi^2$ = 110.19, $p$ < 0.05). M had the highest cone germination potential with a mean of 107 seeds germinating per cone, while T presented the lowest value, with an average of 72 germinated seeds by a cone. Comparatively, T and PV showed 68% and 72%, respectively, of the cone germination potential found in M.

**Table 3.** Factors affecting seed germination of maritime pine seeds from the Mira (M), Tocha (T) and Praia da Vieira (PV) populations obtained by applying a binomial generalized linear model (GLM) to the observed data. Population factor tested using M as default. Statistically significant effects ($p$ < 0.05) on seed germination are highlighted by the * symbol. The Hosmer and Lemeshow test was applied to evaluate the goodness of fit. The model fits well, and no significant differences were found between the model estimates and the seed germination observed data ($p$-value > 0.05).

| | Estimate | Std. Error | z Value | $p$ (>|z|) |
|---|---|---|---|---|
| Intercept | −0.566251 | 0.367505 | −1.738 | 0.082162 |
| Mean seed mass | 16.277369 | 6.389929 | 2.547 | 0.010855 * |
| Mean cone mass | 0.010359 | 0.002757 | 3.757 | 0.000172 * |
| T population (T) | 0.906588 | 0.189284 | 4.790 | $1.67 \times 10{-6}$ * |
| Goodness of fit test: $\chi^2$ = 8.47; df = 8 | | | | 0.39 |

During the first seven months following seed germination, the seedlings' mortality was negligible in all populations. Significant differences were found between the total height ($\chi^2$ = 10.931, $p$ < 0.05) and the stem height ($\chi^2$ = 9.8295, $p$ < 0.05) of the seedlings in distinct populations (Table 4), particularly between PV and the other two populations; seedlings from PV were smaller (lower total and stem height) than those from M and T. No statistical differences were found between the basal stem diameter of the seedlings from the three populations.

**Table 4.** Mean values of **the** total height (T height), stem height (S height), basal shoot diameter (bsd) and the ratio of total height and basal shoot diameter (ratio) of 7-month-old maritime pine seedlings from Mira (M), Tocha (T) and Praia da Vieira (PV) populations. Error (±) indicates standard deviation. Different letters represent statistically significant differences between populations.

| Population | T Height (cm) | S Height (cm) | bsd (mm) | Ratio T Height/bsd |
|---|---|---|---|---|
| M | 21.34 ± 2.94 [a] | 18.18 ± 3.01[a] | 3.52 ± 0.37 [a] | 6.11 ± 0.95 [a] |
| T | 21.06 ± 3.01 [a] | 18.12 ± 3.11[a] | 3.51 ± 0.46 [a] | 5.69 ± 1.09 [a] |
| PV | 19.66 ± 2.90 [b] | 16.71 ± 2.76 [b] | 3.52 ± 0.37 [a] | 6.02 ± 0.88 [a] |

Correlations between seed mass and seedling height and between seed mass and basal shoot diameter were investigated. Although no significant correlation was found between biometric variables and seed mass in seedlings from the 3 studied populations, there was a suggested overall weak positive correlation ($\rho$ = 0.3485, $p$ > 0.05) between basal shoot diameter and seed weight.

## 4. Discussion

This study presents a comparison of reproductive traits between three maritime pine populations in the coastal center of Portugal. An individual approach from cone to seed and from seed to seedling was followed, making it possible to explore direct and relevant correlations underlying the initial stages of maritime pine recruitment.

Cone production is highly variable within maritime pine species and populations, being very sensitive to site conditions and climate fluctuations [11,12]. Results found in the present study indicated that cone mass varied among the three Portuguese maritime pine

populations. The cones collected in M were heavier than those found in T and PV but still lighter than the cones evaluated in other published studies [16,22]. These differences may not express a characteristic of the populations but further the effect of climate conditions during the years prior to cone maturation [20]. The formation of the pine cones used in this work would have occurred during 2017, as cone maturation usually takes about two years [38]. The year 2017 was an extremely hot and dry year in Portugal [35]. The dry weather and hot temperatures lead to a general increase in drought stress in maritime pine stands [39], probably affecting cone production in number and size [20].

Despite the lower mass of pine cones, the mean number of seeds per cone was similar to other studies in Portuguese stands [16] but quite variable between populations. Seed production is not usually a limiting factor in maritime pine recruitment [12], but our results demonstrated a positive correlation between cone mass and the number of seeds. In accordance, the M population had a higher number of seeds and heavier cones. Nevertheless, the cone mass did not explain the low number of seeds obtained in T. Justification may be the observed cone asymmetry in some samples due to the maturation of only part of the seed set.

Seed number and seed mass are often considered good indicators of seed quality [21,22]. Natural selection of heavier seeds can be considered a plant strategy to assure the development of viable embryos capable of effectively germinating and originating more vigorous seedlings [23–25].

The correlation between cone mass and mean seed mass was positively weak, demonstrating the tendency of heavier cones to produce heavier seeds, particularly in the M population. The number of resources allocated to seeds is not dependent on cone size, being genetically controlled by the progenitor tree [21]. In addition to genetic control, the seed mass is also highly dependent on environmental conditions, with increased variability under stress conditions [22]. The seed mass may have been affected by the dry climate registered during 2017, the year of initial cone formation [35]. The values of seed mass found in the present study were comparable to those reported by other authors [16,17] in maritime pine populations, even despite their high variability in populations. The drought stress felt during 2017 affected seed mass and the number of mature seeds. High variances in seed mass were also found by other authors in maritime pine populations from Spain [3,22], Morocco [17,40] and Portugal [16], mostly justified by the local environmental conditions during seed maturation [17,21,38]. As seed mass is a variable and heritable trait actively affected by selective environmental pressures imposed by the environmental conditions [22], the great variability found in seed mass may be justified by the heterogenous characteristics of the Portuguese forest [16], where trees within the same population may be submitted to different abiotic stressors that will consequently affect resource allocation to cones and seeds.

All three populations had germination rates around 52 to 64% 2 months after the beginning of the experiment. These results highlighted one of the main differences found between the coastal Portuguese populations and other Iberian maritime pine populations, where the germination rate was higher than 90% in 1-month germination without pretreatments [4,41]. Higher levels of seed germination were only obtained at the end of the third month, with the highest value reported in the T population (89%). The higher dormancy of seeds from these maritime pine populations may have been a beneficial evolutionary step resulting from their adaptation to specific local conditions or more stressful environments [38].

Studies in Spanish maritime pine populations found that heavier seeds tend to have higher germination rates, with lower germination times [4,41,42]. Such a correlation was very weak in maritime pine populations from Morocco [17,40]. Additionally, a weak, but negative correlation, was found between seed mass and germination capacity by Corticeiro et al. (submitted) in 8 maritime pine populations from Portugal. Results from the current study demonstrated that seed germination was highly related to seed mass and, to a less extent, to cone mass. Although these results seem to contradict data from the pre-

vious study, it involved only 3 of the 8 maritime pine populations used by Corticeiro et al. (submitted). Moreover, determining the mass and testing the germination of each seed individually allowed us to diminish intra-replicate uncertainty. Of those 3 populations, only T exhibited the influence of environment local factors on seed germination capacity. As the relation between seed mass and seed germination rate and timing found across different works has been slightly inconsistent [4,17,26,40–42], recruitment estimates based on seed mass will be more likely to fail due to evident genetic x environment interactions. In addition to lab characterization of the reproductive traits in maritime pine populations, detailed information on local variations of environmental and climate factors is of crucial importance.

Seed mass was positively correlated to seedling biometry, more specifically to the basal shoot diameter of the seedling. However, the overall results suggested that, under controlled conditions, seedling viability was not compromised by seed mass within the range reported in the current study.

## 5. Conclusions

Overall, the results demonstrated that heavier cones tend to generate more mature seeds; however, that may not necessarily originate heavier seeds or seeds with higher germination rates. The M population may potentially have a higher number of seedlings in the field, not due to a higher germination rate but as a consequence of the higher number of seeds per cone. Heavier seeds also tend to originate more robust seedlings, at least during the plant's early stages and under optimal conditions.

Moreover, the current work highlighted the great variability within each population with regards to the studied variables. Such levels of variability, like those reported in other works [11,12,17,18,22], reflect the ecology of maritime pine and its adaptation to distinct environmental conditions, particularly by producing high levels of viable, genetically diverse seeds even under dry conditions.

**Author Contributions:** Conceptualization, P.M. and S.C.; methodology, P.M.; formal analysis, S.C.; investigation, P.M. and S.C.; resources, P.M.; writing—original draft preparation, P.M.; writing—review and editing, S.C.; visualization, S.C.; supervision, P.M.; project administration, P.M.; funding acquisition, P.M. All authors have read and agreed to the published version of the manuscript.

**Funding:** This work was supported by SuSPiRe (PTDC/ASP-SIL/30983/2017), funded by FCT, through COMPETE2020—Programa Operacional Competitividade e Internacionalização (POCI). We acknowledge financial support to CESAM by FCT/MCTES (UIDP/50017/2020+UIDB/50017/2020+ LA/P/0094/2020) through national funds.

**Institutional Review Board Statement:** Not applicable.

**Informed Consent Statement:** Not applicable.

**Data Availability Statement:** Not applicable.

**Acknowledgments:** The authors thank José Cunha and Diana Rodrigues for the extraordinary support on field and lab activities, and to IPMA for the climate data used in this study. Thanks are due to three anonymous reviewers for suggesting important improvements to the original manuscript.

**Conflicts of Interest:** The authors declare no conflict of interest.

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
