# Peer review of "From Cone to Seed and Seedling—Characterization of Three Portuguese Pinus pinaster Aiton Populations"

_2674-1024, doi:10.3390/seeds1020011_

Round 1

Reviewer 1 Report

Manuscript Number:  seeds-1639220

Title:   From cone to seed and seedling – characterization of three Portuguese Pinus pinaster Aiton populations

Authors:  Paula Maia, et al.

Remarks:

This submitted paper investigated variations in cone, seed and seedling characteristics among three populations of Maritime pine, and the authors found that the intra-population variations were high and seed characteristics affected the seedling performance. I consider that the insights obtained from the study should be important for facilitating genetic and morphological diversity maintenances of the species. So, I consider that the content of this paper itself is potentially acceptable for the publication in this journal.

However, the present version of the paper has some major problems; one of them is lack of important information on methodology such as sample numbers and sampling years. Information on the statistical procedure is also insufficient (see comments below in detail), which should be reconsidered carefully. Word uses are often unclear and incorrect throughout the paper, and I could not understand what the authors mean by some sentences.

As a result, I recommend the authors to reconsider again the entire manuscript with a sufficient effort, based on the major and minor suggestions commented as below.

Major Comments:

  1. As commented in the Remarks, the sampling year and sample numbers of cones are unclear. For example, when the cones are sampled? How may numbers of trees per population and cones per tree are constituted in the total of 30 cones per population (as explained in lines 116-)? Such unclearness of the material information may lead to the irrelevance of the study design by the readers, so the author should check again throughout the manuscript and take care for providing the material and sampling information.
  1. Some statistical procedures and explanations are also unclear.

In lines 135-, differences of mother trees in germination rate are not considered, and the readers cannot see whether the difference is brought by the populations or the mother trees within populations. Here the nested linear (mixed) model and the nested-ANOVA, including the effects of both populations and mothers within populations, should be incorporated to explain accurately the variances by the populations. Please reconsider carefully.

In Table 4, I do not understand the effect of “PV population”. I think that this effect is unnecessary and it is enough to consider the effect of “T population”. Please reconsider.

  1. Some results, Figure and Table presentations are unclear.

For example, what error the letter “± xxx” indicate in Tables 2, 3 and 5? In Figure 2, what does the different kind of points indicate? Does one point indicate one tree? or one cone? Such unclearness of presentations may lead to the irrelevance of the study by the readers, so the author should check throughout Tables and Figures once again and take care for providing the clear information.

In lines 234-236, data are not shown. Relationship between seed characteristics and seedling performance should be one of the main theme of the present study, so I suggest to represent the data and statistical information explicitly. Please consider.

  1. There are many incorrect and unclear word uses throughout the manuscript. I inserted word corrections in the printed paper and scanned (as in an attached PDF). So including these points, the authors should check once again throughout the manuscript and improve before the resubmission.

Specific comments:

Introduction

  1. Lines 43-: “weather (-> climate) conditions”. What climate? Explain more concretely.
  2. Lines 71-75: “Three maritime pine populations …”. These explanations should be moved to the Materials & Methods section.

Materials and Methods

  1. Table 1. This information is already summarized in the text and also visualized in Figure 1. This table is unnecessary.

Results

  1. Figure 2. Variables explained in the legend (seed weight and germination rate) do not correspond with those in the figure (cone weight vs. seed weight). Please correct.
  2. Table 5. Text explains that basal shoot diameter differs significantly among populations (lines 227-), but not significant in the column “bsd (mm)” in this table. Please correct.

Discussion

  1. Lines 312-313: “However, the overall results suggested …”. I do not understand how does this sentence correspond with the former sentence “Heavier seeds tended to …” (lines 311-)? Please explain more clearly.
  2. Lines 324-. “viable seeds” should be “(genetically) diverse seeds”. Please reconsider.

Reviewer 2 Report

In the present work, the mass of cones and seeds of three populations of maritime pine from the Coastal Center of Portugal are determined and compared, and the relationships between cone and seed traits with germination phenology and initial growth of seedlings are analyzed. The results demonstrated that seed weight is a good indicator of robust seedlings and reinforce the great intrapopulation variability of maritime pine and its ability to successfully adapt to diverse environments.
We consider that the work is adequately justified and only some aspects should be improved, from our point of view:
- lines 123 to 133. Explain in more detail the procedure for determining the speed of germination per cone.
- Figure 2 does not detail the meaning of the symbols used in the correlation, and it would be desirable to include a more detailed explanation than the one offered (lines 178-183).
- Table 4 must include the meaning of all the symbols used.
- Table 5 must specify the meaning of the different letters used (statistics).

Reviewer 3 Report

Line 2 Title – please, write Pinus pinaster using italic

Line 96 – Climate data – please add references “…was obtained from IPMA (2019, 2018, 2017) [36-38]

Line 132 – Seedling nursing and biometric traits – please explain exactly how seedlings were measured? From where did the measurement start? From shoot basal? In table 5. are results of S height – section Material and methods does not mention this parameter.

Table. 4 – is no results of M population (M)

Table 5. line 231 – what “steam height” means? Shoot height or stem height?

Line 229 – in case of basal shoot diameter were no differences between populations. Differences were in T height and S height.

References – should be corrected in accordance with editorial requirements.

Round 2

Reviewer 1 Report

Manuscript Number:  seeds-1639220_v2

Title:   From cone to seed and seedling – characterization of three Portuguese Pinus pinaster Aiton populations

Authors:  Paula Maia, et al.

Remarks:

This resubmitted paper has improved mostly, such as information on methodology and statistical procedure, and also writing styles and word uses. I consider that the revised version has become close to be acceptable. However, some important information and Table/Figure presentation have not been still updated, and some word uses which I commented before have not been still corrected. The authors should reconsider the major and specific points I suggested below, and check again throughout the manuscript before the resubmission.

Major comments:

1. The authors have revised information on the sampling year and sample numbers of cones. I understand that three cones per tree were sampled and one out of them were processed to the measurements of cone, seed and seedling characteristics. However, if so, I do not still understand how the authors did with the other two cones. Weren’t the two cones used to any experiments? If so, this is the same as that the authors sampled one cone per tree, and therefore the authors should write so. Please explain the fact explicitly.

2. At the time of draft-review, I suggested the authors to explain about the error letter “±” in some tables. The authors revised to explain in Table 2 (the present version), but not in Tables 1 and 4 (the present version) yet. Including this point, please check the presentation of results, tables and figures once again to explain the data carefully.

Specific comments:

Introduction

3. The sentence of Line 70-71 “Three maritime pine populations …” is redundant of Line 74-76. This can be cut.

Materials and Methods

4. Line 105-. “the pine cone” should be “the cone”.

5. Line 106-. This sentence “Each seed used in the …” should be checked again and corrected. The words “in the germination test” are duplicated.

Results

6. Line 230-. “p>0.05” should be “p<0.05”?

7. Line 230-. In this sentence, there is no explanation on the difference of “bsd”.
